# Red blood cell indices versus serum ferritin as surrogate markers of iron deficiency during pregnancy

Ochuwa Adiketu Babah[1,2,3]*, Chisom Florence Chieme[3], Ajibola Ibraheem Abioye[4], Samuel Olusegun Spaine[5], Titilope Adenike Adeyemo[3,6], Bosede Bukola Afolabi[1,2,3]

1 Department of Global Public Health, Karolinska Institutet, Solna, Stockholm, Sweden, 2 Department of Obstetrics and Gynaecology, Faculty of Clinical Sciences, College of Medicine, University of Lagos/ Lagos University Teaching Hospital, Idi-Araba, Lagos State, Nigeria, 3 Centre for Clinical Trials and Implementation Science (CCTRIS), College of Medicine, University of Lagos, Idi-Araba, Lagos, Nigeria, 4 Department of Global Health and Population, Harvard T.H. Chan School of Public Health, Boston, Massachusetts, United States of America, 5 Sheikh Muhammad Jiddah General Hospital, Kano, Kano State, Nigeria, 6 Department of Haematology and Blood Transfusion, Faculty of Clinical Sciences, College of Medicine, University of Lagos/Lagos University Teaching Hospital, Idi-Araba, Lagos State, Nigeria

* ochuwa.babah@ki.se, obabah@unilag.edu.ng

## Abstract

### Background

Serum ferritin testing is the most commonly used method for screening for iron deficiency. However, iron deficiency screening is not routinely done in low-middle-income countries, including Nigeria, often due to the cost of laboratory evaluation.

### Aim

This study determined the diagnostic value of red blood cell indices, which are cheaper and quicker to conduct, compared to serum ferritin to diagnose iron deficiency during pregnancy.

### Methodology

A cross-sectional study of 857 pregnant women at 36 weeks gestation in Nigeria. Standard laboratory techniques assayed mean corpuscular volume (MCV), mean cell haemoglobin (MCH), mean cell haemoglobin concentration (MCHC), red cell distribution width (RDW) and serum ferritin. Spearman's rank correlation coefficient of each of the complete blood count parameters (MCV, MCH, MCHC and RDW) with serum ferritin were assessed and their diagnostic accuracy relative to iron deficiency (defined as ferritin <30ng/mL) was evaluated.

**Data availability statement:** The dataset analysed for this study can be accessed on Open Science Framework at: https://osf.io/enx7z.

**Funding:** Bill & Melinda Gates Foundation (Investment ID INV-017271) Grant for IVON TRIAL, the research in which this study was nested.

**Competing interests:** The authors have declared that no competing interests exist.

## Results

Mean age of the pregnant women was 27.7±5.8 years. Median (IQR) was 10.3 (IQR: 9.6–11.0) g/dL for haemoglobin, and 84.0 (IQR: 47.0–157.9) ng/mL for ferritin. Serum ferritin levels have significant correlation with RDW, r=−0.12, p<0.001 and MCH, r=0.10, p=0.003; but not with MCV, r=0.06, p=0.083 and MCHC, r=0.04, p=0.293. RDW was found to be the best discriminator for iron deficiency based on area under curve (AUC) 59.9% (95%CI: 56.6% – 63.2%), sensitivity 65.6% and specificity 53.8% at best cut-off 14.7fL. On restricting analysis to those with anaemia, the findings did not change materially.

## Conclusion

The diagnostic value of red blood cell indices, compared to serum ferritin, in detecting iron deficiency and iron deficiency anaemia is poor and should not play a role in diagnosing iron deficiency in pregnancy in a low-resource setting.

## Introduction

Globally, anaemia affected 37% of pregnant women in the reproductive age group in 2019 [1]. The highest burden of anaemia is observed in low-middle-income countries like Nigeria [1]. In Nigeria, the prevalence of anaemia is 58% among pregnant women in the reproductive age group [2]. Although the cause of anaemia is multifactorial, it has been largely attributed to insufficient dietary intake of iron which results in iron deficiency [1]. Low iron stores in pregnancy have been associated with a number of unfavourable outcomes, such as anaemia in the mother, preterm birth, low birth weight, and enduring cognitive impairments in the offspring [3,4].

Due to the widespread incidence of iron deficiency anaemia (IDA), haemoglobin concentration has historically been the preferred technique for determining iron status [5]. Its low cost, easy measurement, and wide availability make it very useful, especially in environments with limited resources; however, there are issues with depending just on haemoglobin screening [6]. Despite physiological tissue-level consequences, anaemia, as the endpoint of negative iron balance, lacks the sensitivity to identify early stages of iron insufficiency. Therefore, diagnosing iron deficiency in pregnancy cannot be done with enough sensitivity or specificity based on haemoglobin concentration alone.

Serum ferritin, total iron binding capacity, serum transferrin, transferrin saturation, and hepcidin are examples of iron-specific biomarkers that can be used to differentiate iron deficiency anaemia from other types of anaemia [6]. Serum ferritin is the earliest indicator of decreasing iron stores and remains unaffected by recent iron intake [7]. Widely regarded as the top choice for assessing iron deficiency in pregnancy, serum ferritin is considered the most dependable initial test for diagnosing absolute iron deficiency, despite its susceptibility to elevation during active infection or inflammation [8,9].The serum ferritin level serves as the most sensitive and specific test

for identifying iron deficiency, typically indicated by a level below 15 ng/mL, with a higher cut-off below 30 ng/mL recommended in settings of inflammation [5,10]. Hepcidin has also been found to have good diagnostic accuracy for diagnosis of iron deficiency and iron deficiency anaemia, but its use is limited by cost when compared to the cost of other tests [11,12]. The ideal determinant for body's iron status is bone marrow iron; however, bone marrow biopsies are invasive, costly, and not commonly performed during pregnancy [13].

In low-middle-income countries like Nigeria, routine screening for iron deficiency is not commonly practiced, primarily due to the associated costs of laboratory evaluations [14]. This lack of widespread screening poses significant challenges in identifying and managing iron deficiency, particularly during pregnancy, a critical period when maternal and foetal health are intricately linked. Complete blood count is cheap in Nigeria compared to serum ferritin.

Recognizing the importance of accurately diagnosing iron deficiency anaemia in pregnancy, this study endeavours to compare the diagnostic value of various red blood cell indices with serum ferritin serving as the established reference for diagnosing iron deficiency during pregnancy [15]. Previous studies like those by Abioye and colleagues did not assess other components of complete blood count, apart from haemoglobin concentration [11]. Our study aimed to provide evidence on the correlation between serum ferritin and red blood cell indices such as red blood cell distribution width (RDW), mean cell volume (MCV), mean cell haemoglobin concentration (MCHC), and mean cell haemoglobin (MCH). It also determined the predictive value of the red blood cell indices in predicting iron deficiency during pregnancy, using serum ferritin as the reference.

## Materials and methods

### Study design

A cross-sectional study nested within the IVON trial. The IVON trial was a randomized controlled trial which evaluated the effectiveness and safety of intravenous ferric carboxymaltose versus oral ferrous sulphate for treatment of iron deficiency anaemia in pregnancy [16].The trial enrolled 1,056 pregnant women with moderate or severe anaemia at 20–32 weeks gestational age and treated them with either intravenous ferric carboxymaltose and oral ferrous sulphate and compared the effectiveness and safety of both medications. The participants were followed up till six weeks postpartum. Though the clinical trial assessed women at five time points we choose to conduct this analytic cross-sectional study at the 36-week gestation time point because the sample size for the clinical trial was estimated from data obtained at a 36 weeks' time point [16].

### Study setting

The clinical trial enrolled participants from 10/08/2021–15/12/2022 from eleven public healthcare facilities in Lagos and Kano states of Nigeria. The health facilities comprised five primary, four secondary, and two tertiary health facilities.

### Participants

The participants for this study were between the ages of 15–49 years, presenting for antenatal care follow-up at 36 weeks gestational age and who had their haemoglobin concentration and serum ferritin results reported. Excluded from the clinical trial were women who had haemoglobinopathies like sickle cell disorders, previous hypersensitivity to iron preparations, human immunodeficiency virus infection, severe malabsorption syndrome, and autoimmune disease.

### Sample size determination

Using Cochrane's formula [17], a sample size of 414 was calculated to be adequate for this study based on a prevalence of 41.2% for iron deficiency among anaemic pregnant women in Nigeria [18], at a 5% significance level and adjusting for 10% attrition.

## Data collection

The pregnant women had their sociodemographic and obstetric data collected at enrolment into the clinical trial by interviewer-administered questionnaires. Research physicians and nurses conducted detailed clinical assessments at each clinic visit. All participants received standard antenatal care as per Nigerian national guidelines, with antenatal visits every 1–4 weeks depending on gestational age, intermittent preventive treatment twice for malaria, daily 5 mg folic acid and 100 mg of vitamin C three times daily until the end of pregnancy. At their 36 weeks follow-up visit, their blood specimens were assayed for complete blood count and iron panel which included serum ferritin assay. All laboratory assays were conducted at Synlab Laboratories Limited, an internationally accredited laboratory in Lagos, Nigeria. Complete blood count was done using haematology autoanalyzer (SYSMEX XN-L series, Japan) while serum ferritin assay was done using ARCHITECT Ferritin assay method.

## Outcome measures

The predictors of interest were red blood cell indices comprising RDW, MCV, MCHC, and MCH. The outcomes of interest were iron deficiency defined as serum ferritin <30ng/mL, and iron deficiency anaemia defined as concurrent haemoglobin concentration < 11g/dL and serum ferritin <30ng/mL.

## Data analysis

Data was analysed with Stata version 18.0 (Stata Corp. 2023. Stata Statistical Software: Release 18. College Station, TX: Stata Corp LLC).Continuous variables were summarized and presented as mean (standard deviation) or median (interquartile range) depending on the population distribution on normality testing with Kolmogorov Smirnov test. Categorical variables were presented as frequencies and percentage. Spearman's rank correlation coefficient was used to determine correlations between serum ferritin versus each red blood index. Receiver operating characteristic (ROC) curve was constructed to determine the predictive value for iron deficiency of each of the red blood cell indices – RDW, MCV, MCHC, MCH. Area under the curve (AUC) was reported for each haematological index. Sensitivity and specificity at optimal threshold were determined using Youden's index [19]. The same analyses were repeated in a subgroup of the pregnant women with iron deficiency anaemia. Threshold for statistical significance was set at p-value of <0.05. No adjustment for missingness was necessary and complete case analysis was done.

## Ethics

The clinical trial was conducted with strict adherence to the principles of research ethics as outlined in the Declaration of Helsinki. Ethical approvals were obtained from the National Health Research Ethics Committees (NHREC/01/01/2007-04/02/2021); Aminu Kano Teaching Hospital (NHREC/28/01/2020/AKTH/EC/2955) and the State Ministry of Health (MOH/Off/797/T.1/2102) in Kano State; and Lagos University Teaching Hospital HREC (ADM/DCST/HREC/APP/3971) in Lagos State. Administrative approvals were obtained from the Primary Health Care Board (LS/PHCB/MS/1128/VOL.VII/100) and Health Service Commission (LSHSC/2222/VOLIII) in Lagos State. All the study participants signed written informed consent before participating in the IVON TRIAL. Eligible pregnant women below 18 years were considered emancipated and able to give written consent themselves based on national guideline that encourages inclusivity in research [20]. Confidentiality of data was ensured. The data released for this study was anonymized.

## Results

Of the 1,056 participants in the clinical trial, 189 did not have both haemoglobin concentration and serum ferritin reported, and five each had missing values for haemoglobin concentration or serum ferritin only, leaving 857 eligible pregnant women for this study. Fig 1 is a summary chart of participants inclusion in this study.

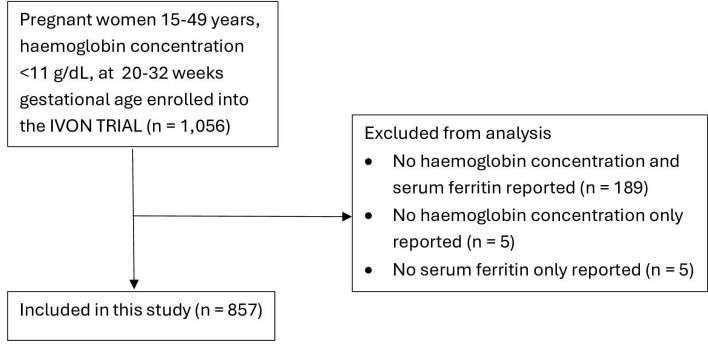

**Fig 1. Participant's inclusion in this study.**

The mean age of the pregnant women was 27.7±5.8 years, with median parity of 1 (IQR: 0–3). Their median haemoglobin concentration was 10.3 (IQR: 9.6–11.0) g/dL, and median serum ferritin level 84.0 (IQR: 47.0–157.9) ng/mL. Table 1 provides detailed information about the demographic and clinical profile of the study participants.

Of the complete blood count indices, RDW and MCH had significant correlation with serum ferritin, $r = -0.12$, $p < 0.001$ and $r = 0.10$, $p = 0.003$ respectively, while MCV and MCHC do not have a significant correlation with serum ferritin $r = 0.06$, $p = 0.083$ and $r = 0.04$, $p = 0.293$ respectively, Fig 2.

**Table 1. Sociodemographic and clinical profile of the study participants (n = 857).**

| Sociodemographic/clinical characteristic | Mean±SD |
|---|---|
| Age (years) | 27.7±5.8 |
| | **Median (IQR)** |
| Parity | 1 (0–3) |
| Haemoglobin concentration (g/dL) | 10.3 (9.6–11.0) |
| Serum ferritin (ng/mL) | 84.0 (47.0–157.9) |
| Mean corpuscular volume (fL) | 89.9 (84.2–96.4) |
| Mean cell haemoglobin (pg/cell) | 28.6 (26.8–30.5) |
| Mean cell haemoglobin concentration (g/dL) | 32.0 (30.0–33.2) |
| Red blood cell distribution width (fL) | 14.6 (13.6–15.8) |
| | **Frequency (%)** |
| Marital Status | |
| Married | 814 (95.0) |
| Single | 31 (3.6) |
| Cohabiting | 9 (1.1) |
| Divorced/separated | 3 (0.35) |
| Place of residence | |
| Urban | 781 (91.1) |
| Rural | 76 (8.9) |
| Highest level of education | |
| Completed Tertiary | 266 (31.0) |
| Completed Secondary | 438 (51.1) |
| Completed Primary | 106 (12.4) |
| No formal education | 47 (5.5) |

SD – standard deviation, IQR – interquartile range.

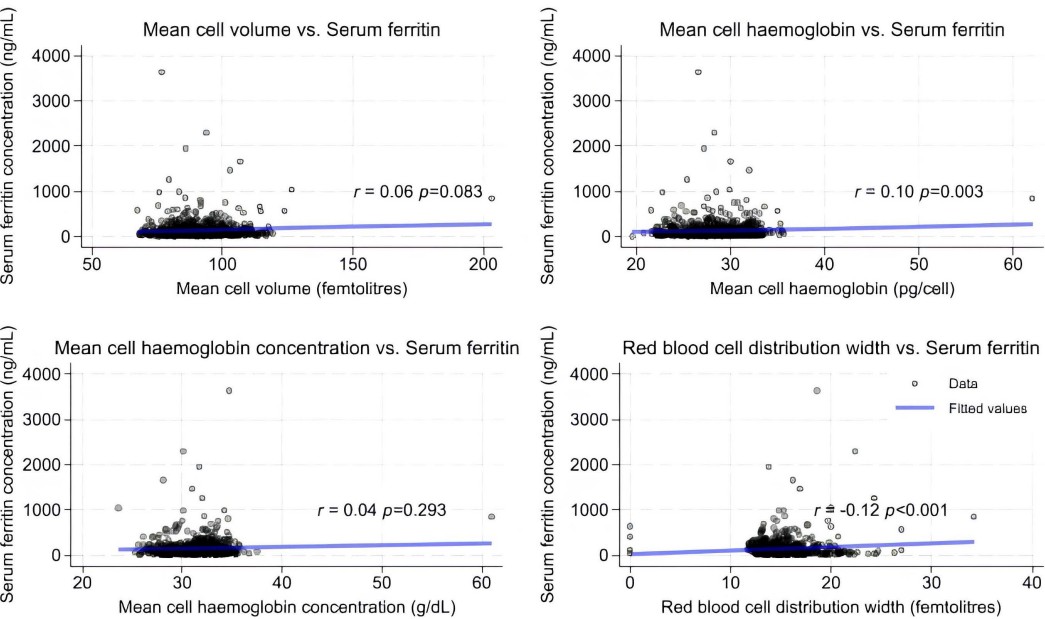

**Fig 2. Scatter charts showing correlation of complete blood count parameters with serum ferritin.** *r = correlation coefficient.*

We compared the diagnostic accuracy of using each of these haematological indices for diagnosis of iron deficiency during pregnancy (Fig 3). In the entire sample of pregnant women, we found that of the red cell indices, the best discriminator for iron deficiency was RDW with an AUC = 59.9% (95%CI: 56.6% – 63.2%), sensitivity 65.6% and specificity 53.8%. The optimal cut-off for RDW was 14.7fL. Other complete blood count components had poor discriminatory capability with AUC = 49.0% for MCV, AUC = 41.5% for MCH and AUC = 40.9% for MCHC.

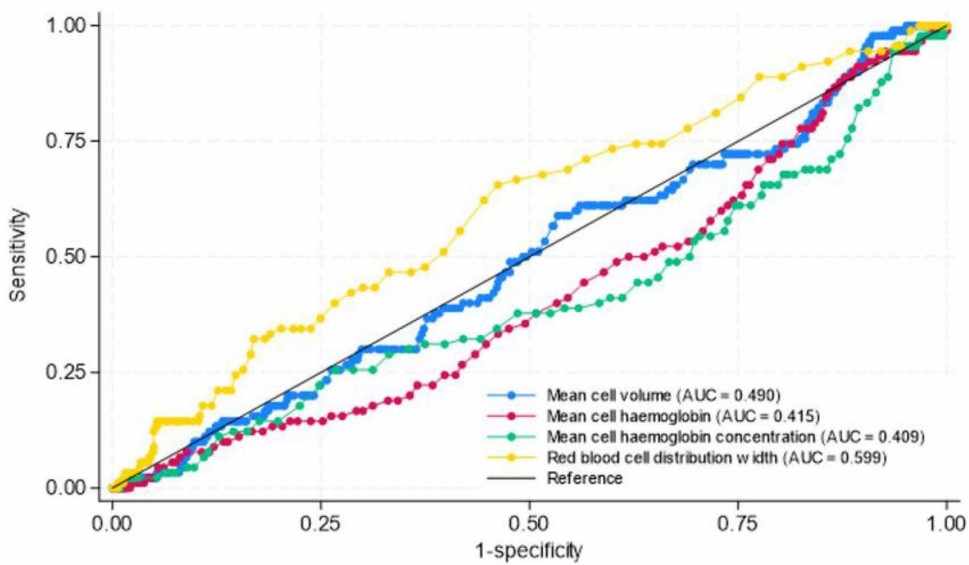

**Fig 3. Diagnostic accuracy of complete blood count parameters for diagnosis of IRON DEFICIENCY during pregnancy using serum ferritin as reference (n = 857).** *AUC refers to area under receiver operation characteristic curve.*

Next, we repeated the analysis on subgroup of pregnant women with anaemia only. In the subgroup of pregnant women with anaemia, we determined the diagnostic value of using these methods for diagnosis of iron deficiency anaemia. The findings are presented in Fig 4. Of the red cell indices, best discriminator for iron deficiency was RDW with AUC = 58.9% (95%CI: 55.0% – 62.8%), sensitivity 64.7% and specificity 53.5% at best cut off 14.7fL. Other components of complete blood count had AUC = 51.6% for MCV, AUC = 44.0% for MCH, and AUC = 40.2% for MCHC.

## Discussion

This study evaluated the predictive value of red blood cell indices for diagnosing iron deficiency during pregnancy using serum ferritin as the reference. Though RDW and MCH have significant correlation with serum ferritin, MCV and MCHC do not. Though RDW had the best predictive value compared to MCH, MCV and MCHC, none of the complete blood count parameters evaluated had good enough discriminatory power for diagnosing iron deficiency and iron deficiency anaemia during pregnancy.

Ferritin is an intracellular protein which stores in the body cells; it is thus used to directly assess iron stores in the body. The red blood cell indices indirectly assess iron content of the body by determining the quality of the red blood cells in terms of haemoglobin content or the size of the cells. Haemoglobin contains a Heme and a globin moiety. Heme contains iron, depicting a crucial role of iron in erythropoiesis. This study has shown indirect assessment of iron content of the body using red blood cell indices like MCH. MCHC, MCV and RDW to diagnose iron deficiency during pregnant to be inferior to the use of serum ferritin.

Like our study, an earlier study in Oxford, United Kingdom found the predictive value of red blood cell indices for diagnosing iron deficiency in pregnancy to be poor [21]. Though this study did not evaluate the diagnostic value of using RDW for this purpose like we did, it evaluated other parameters like haematocrit and red blood cell count. Both studies evaluated the diagnostic value of MCV, MCH and MCHC. Current study found RDW to be the best predictor of iron deficiency in pregnancy compared to other red blood cell indices. On the contrary, a similar study on non-pregnant Filipino women found red blood cell count, haematocrit, MCV, MCH, and MCHC to have an acceptable predictive value (AUC above 70%)

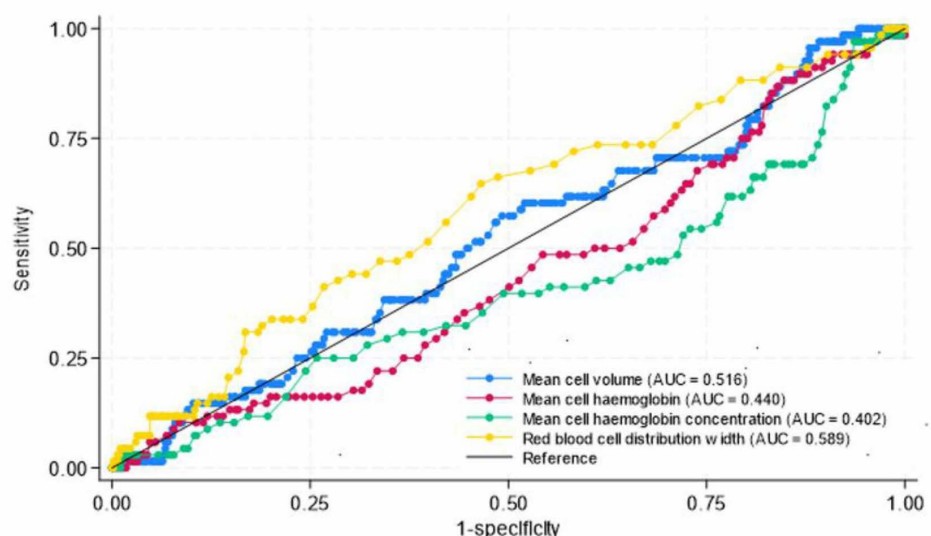

**Fig 4. Diagnostic accuracy of complete blood count parameters for diagnosis of IRON DEFICIENCY ANAEMIA during pregnancy using serum ferritin as gold standard (n = 634).** *AUC refers to area under receiver operation characteristic curve.*

for diagnosing iron deficiency [22]. Like haemoglobin concentration, RDW may be affected by altitude. In addition, it may be altered in nutritional deficiency states like vitamin B12 and folate deficiencies; normal levels may occur in haemoglobinopathies [23]. It is important to take into consideration these factors when interpreting the results.

Regarding the effect of pregnancy on iron indices, a previous study found a fall in the levels of serum ferritin in the third trimester of pregnancy compared to the levels in first and second trimesters [24]. Alterations in the red blood cell indices also do occur in pregnancy as a result of the physiological haemodilution which aggravates the occurrence of anaemia with peak effect between 32–34 weeks [25]. Considering these findings, it is possible that the haematological changes that occur during pregnancy might be a reason for the observed loss of association between the red blood cell indices and iron deficiency. However, this was a main reason why we examined women at specific time point (36 weeks gestation) in this study. This will be worth exploring in future research especially as our recent survey found that 85% of Nigerian maternal healthcare workers consider using complete blood count for screening for iron deficiency anaemia in pregnancy [14].

Like the current study, a study in India found a significant correlation between RDW and serum ferritin [26]. However, the Indian study found a stronger correlation of 0.420 between RDW-coefficient of variation compared to 0.09 obtained in our study, indicating a higher predictive power of RDW-coefficient of variation in identifying iron deficiency in the Indian population [26]. In addition, the study observed significant correlations in second- and third-trimester anaemic pregnant women (with Hb < 10.5 g/dL) between ferritin and other red cell indices, such as MCV, MCH, MCHC, and RBC count [26]. Our study also found a significant correlation between MCH and serum ferritin, but not with MCV and MCHC; with limited predictive value for these red cell indices for diagnosing iron deficiency or iron deficiency anaemia. The variations in the strength and nature of these associations may stem from differences in population characteristics, geographical factors like altitude, and techniques of measurement of the laboratory markers. Our study evaluated pregnant women in late third trimester only while the Indian study had a mix of pregnant women in second and third trimester of pregnancy. It is known that red cell indices differ across trimesters of pregnancy [27].

Furthermore, a study in Pakistan further supports the relevance of RDW as a valuable diagnostic tool, reporting high sensitivity (86.96%), specificity (85.88%), and an overall diagnostic accuracy of 86.50% for RDW in detecting iron deficiency anaemia when serum ferritin served as the gold standard [28]. The predictive accuracy observed in the Pakistani study reinforces our findings on RDW's utility in diagnosing iron deficiency, highlighting RDW's potential effectiveness as a diagnostic tool across diverse populations. Collectively, these comparisons emphasise the need to tailor diagnostic approaches based on population-specific characteristics, gestational stage, and the chosen biomarkers to accurately identify and manage iron deficiency in pregnancy.

On the importance of combining laboratory measures, a study conducted in the United States on an African American population found that a combination of low Hb (≤9.7 g/dL) and high RDW (≥15fL) at gestational age before 20 weeks effectively predicted iron deficiency with high specificity [29]. The focus on gestational age before 20 weeks in the U.S. study highlights the potential significance of early pregnancy physiological changes in iron metabolism, which differ from our study, where the cohort consisted of women at later gestational ages [29]. These findings suggest that lower Hb levels in conjunction with elevated RDW may be highly predictive in certain demographic and gestational subgroups, indicating that the utility of RDW as a diagnostic marker may be population- and stage-specific. It also emphasizes the need to explore in future research the possibility of using a combination of the red blood cell indices to improve the diagnostic value of using complete blood count parameters for predicting iron deficiency during pregnancy.

In recognising the limitations of haemoglobin alone as an indicator of iron deficiency, our study evaluated red cell indices, such as RDW, MCH, MCHC and MCV, as diagnostic tools for iron deficiency during pregnancy [30,31]. However, a major limitation of this study is the use of serum ferritin as the reference for defining iron deficiency as opposed to bone marrow biopsy because the IVON TRIAL [16], in which our current study is nested, did not conduct bone marrow biopsies which is the gold standard for body for determining body iron. Considering that serum ferritin level is known to be affected

by inflammation, we adjusted for this by using a higher threshold of 30ng/mL for diagnosis of iron deficiency in this study. There is a relative lack of studies evaluating the accuracy of the red blood cell indices for iron deficiency screening during pregnancy. This study provides an insight into the diagnostic value of using the red blood cell indices for this purpose, identified new gaps in knowledge and directions for future research on related subject.

## Conclusion

The findings from this study suggest that the diagnostic value of red blood cell indices- specifically MCV, MCH, MCHC, and RDW, is limited in detecting iron deficiency and iron deficiency anaemia in low-resource settings. Despite some correlations observed between the red blood cell indices and serum ferritin levels in various studies, our data indicate that these measures lack the sensitivity and specificity necessary to reliably identify iron deficiency. Given the constraints of low-resource settings, where advanced diagnostic tools like facilities for serum ferritin assay may be scarce, reliance on these indices alone could lead to underdiagnosis or misclassification of iron deficiency, potentially leaving many cases untreated.

## Acknowledgments

Appreciations to the site coordinators and research nurses who assisted with the research data collection.

## Author contributions

**Conceptualization:** Ochuwa Adiketu Babah, Chisom Florence Chieme, Titilope Adenike Adeyemo, Bosede Bukola Afolabi.

**Data curation:** Ochuwa Adiketu Babah, Samuel Olusegun Spaine.

**Formal analysis:** Ochuwa Adiketu Babah, Chisom Florence Chieme, Ajibola Ibraheem Abioye.

**Funding acquisition:** Ochuwa Adiketu Babah, Bosede Bukola Afolabi.

**Methodology:** Ochuwa Adiketu Babah, Ajibola Ibraheem Abioye, Samuel Olusegun Spaine, Titilope Adenike Adeyemo, Bosede Bukola Afolabi.

**Project administration:** Ochuwa Adiketu Babah, Chisom Florence Chieme, Titilope Adenike Adeyemo, Bosede Bukola Afolabi.

**Software:** Ochuwa Adiketu Babah.

**Supervision:** Bosede Bukola Afolabi.

**Writing – original draft:** Ochuwa Adiketu Babah, Chisom Florence Chieme.

**Writing – review & editing:** Ochuwa Adiketu Babah, Ajibola Ibraheem Abioye, Samuel Olusegun Spaine, Titilope Adenike Adeyemo, Bosede Bukola Afolabi.

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
