## [Decision Letter · Decision Letter 0]

12 Jul 2025

PONE-D-25-15616
Red blood cell indices versus serum ferritin as surrogate markers of iron deficiency during pregnancy
PLOS ONE

Dear Dr. Babah, 

Thank you for submitting your manuscript to PLOS ONE. After careful consideration, we feel that it has merit but does not fully meet PLOS ONE’s publication criteria as it currently stands. Therefore, we invite you to submit a revised version of the manuscript that addresses the points raised during the review process.

We look forward to receiving your revised manuscript.

Kind regards,

Oluyinka Ajibola Iyiola

Academic Editor

PLOS ONE

Journal Requirements:

Bill & Melinda Gates Foundation (Investment ID INV-017271) Grant for IVON TRIAL, the research in which this study was nested.

Reviewers' comments:

Reviewer's Responses to Questions

**Comments to the Author**

1. Is the manuscript technically sound, and do the data support the conclusions?

Reviewer #1: Yes

Reviewer #2: Yes

2. Has the statistical analysis been performed appropriately and rigorously? 

Reviewer #1: Yes

Reviewer #2: Yes

3. Have the authors made all data underlying the findings in their manuscript fully available?

Reviewer #1: Yes

Reviewer #2: Yes

4. Is the manuscript presented in an intelligible fashion and written in standard English?

Reviewer #1: Yes

Reviewer #2: Yes

5. Review Comments to the Author

Reviewer #1: A paragraph incorporating the role of ferritin and red cell indices in Fe metabolism should be added to elucidate the importance of these parameters. An explanation for the main observation that the red cell indices cannot replace ferritin estimation should also be given.

Reviewer #2: -The units (g/dl) for should be g/dL in the abstract, table 1

-In the abstract conclusion, the red blood cels versus serum ferritin should be discussed

- The unit of serum ferritin (ng/mL) should be consistent throughout and not changed to µg/mL at the discussion section

-The first reference in the Reference section should be rewritten by closing up the gaps

6. PLOS authors have the option to publish the peer review history of their article (what does this mean?). If published, this will include your full peer review and any attached files.

Reviewer #1: No

Reviewer #2: **Yes: **Katherine Olabanjo OLUFOLABO

---

## [Author Response · Author response to Decision Letter 1]

21 Jul 2025

PONE-D-25-15616

Red blood cell indices versus serum ferritin as surrogate markers of iron deficiency during pregnancy

PLOS ONE

Dear Dr Oluyinka Ajibola Iyiola,

Academic Editor,

PLOS One.

Thank you for the valuable time you and the reviewers have spent to critically appraise this manuscript. We have addressed the review comments and provided a summary as a point-by-point response below. Page numbers cited are of the manuscript with tracked changes.

Thank you.

Best regards,

Ochuwa A. Babah

Corresponding Author

Journal Requirements

Response

Thank you. The manuscript title page has been formatted according to journal requirements by using Asterix to indicate the corresponding author and by providing only email address for the corresponding author. Page 1.

Response

Thank you. We have rephrased our data sharing statement as follows:

The dataset analysed for this study can be accessed on Open Science Framework at: https://osf.io/enx7z

This statement is now in the manuscript on Page 19.

Bill & Melinda Gates Foundation (Investment ID INV-017271) Grant for IVON TRIAL, the research in which this study was nested.

Response

Done. Now on Page 19 of the manuscript.

Response

Thank you. We have cross checked the manuscript and hereby confirm that this is in order.

Response

Thank you. This is not applicable as none of the reviewers recommended any specific citation.

Response

Thank you. We have checked and confirm our references to be in order.

Reviewer #1

A paragraph incorporating the role of ferritin and red cell indices in Fe metabolism should be added to elucidate the importance of these parameters. An explanation for the main observation that the red cell indices cannot replace ferritin estimation should also be given.

Response

Thank you. We have expanded the discussion section based on the comment above. We added on Page 13, last paragraph that:

Ferritin is an intracellular protein which stores in the body cells; it is thus used to assess iron stores in the body. The red blood cell indices indirectly assess iron content of the body by determining the quality of the red blood cells in terms of haemoglobin content or the size of the cells. Haemoglobin contains a Heme and a globin moiety. Heme contains iron, depicting a crucial role of iron in erythropoiesis. This study has shown indirect assessment of iron content of the body using red blood cell indices like MCH. MCHC, MCV and RDW to diagnose iron deficiency during pregnant to be inferior to the use of serum ferritin.

Reviewer #2

The units (g/dl) for should be g/dL in the abstract, table 1

Response

Thank you. We have replaced on g/dl in the manuscript with g/dL. Abstract Page 2, last paragraph; Page 10, second paragraph; Table 1 on Page 11

In the abstract conclusion, the red blood cells versus serum ferritin should be discussed.

Response

Thank you for the suggestion. Done. Page 3, paragraph 2 (abstract conclusion).

The unit of serum ferritin (ng/mL) should be consistent throughout and not changed to µg/mL at the discussion section.

Response

Thank you. The typographical error has been corrected. Page 17, Paragraph 2.

The first reference in the Reference section should be rewritten by closing up the gaps.

Response

This has been done. Thank you. Page 19, Reference 1.

---

## [Decision Letter · Decision Letter 1]

24 Sep 2025

Red blood cell indices versus serum ferritin as surrogate markers of iron deficiency during pregnancy

PONE-D-25-15616R1

Dear Dr. Babah,

We’re pleased to inform you that your manuscript has been judged scientifically suitable for publication and will be formally accepted for publication once it meets all outstanding technical requirements.

Kind regards,

Tiruneh Adane

Academic Editor

PLOS ONE

Additional Editor Comments (optional):

Reviewer #1:

Reviewer #2:

Reviewers' comments:

Reviewer's Responses to Questions

**Comments to the Author**

1. If the authors have adequately addressed your comments raised in a previous round of review and you feel that this manuscript is now acceptable for publication, you may indicate that here to bypass the “Comments to the Author” section, enter your conflict of interest statement in the “Confidential to Editor” section, and submit your "Accept" recommendation.

Reviewer #1: All comments have been addressed

Reviewer #2: All comments have been addressed

2. Is the manuscript technically sound, and do the data support the conclusions?

Reviewer #1: (No Response)

Reviewer #2: Yes

3. Has the statistical analysis been performed appropriately and rigorously? 

Reviewer #1: (No Response)

Reviewer #2: Yes

4. Have the authors made all data underlying the findings in their manuscript fully available?

Reviewer #1: (No Response)

Reviewer #2: Yes

5. Is the manuscript presented in an intelligible fashion and written in standard English?

Reviewer #1: (No Response)

Reviewer #2: Yes

6. Review Comments to the Author

Reviewer #1: (No Response)

Reviewer #2: The authors had addressed all review comments but there is still the need to close up the gap between fact- and sheets of Reference 1 in the Reference Section

7. PLOS authors have the option to publish the peer review history of their article (what does this mean?). If published, this will include your full peer review and any attached files.

Reviewer #1: No

Reviewer #2: **Yes: **Katherine Olabanjo OLUFOLABO

---

## [Editor Report · Acceptance letter]

PONE-D-25-15616R1

PLOS ONE

Dear Dr. Babah,

I'm pleased to inform you that your manuscript has been deemed suitable for publication in PLOS ONE. Congratulations! Your manuscript is now being handed over to our production team.

Kind regards,

on behalf of

Dr. Tiruneh Adane

Academic Editor

PLOS ONE